# On Characterizing the Capacity of Neural Networks Using Algebraic Topology

## Abstract

The learnability of different neural architectures can be characterized directly by computable measures of data complexity. In this paper, we reframe the problem of architecture selection as understanding how data determines the most expressive and generalizable architectures suited to that data, beyond inductive bias. After suggesting algebraic topology as a measure for data complexity, we show that the power of a network to express the topological complexity of a dataset in its decision boundary is a strictly limiting factor in its ability to generalize. We then provide the first empirical characterization of the topological capacity of neural networks. Our empirical analysis shows that at every level of dataset complexity, neural networks exhibit topological phase transitions and stratification. This observation allowed us to connect existing theory to empirically driven conjectures on the choice of architectures for a single hidden layer neural networks.

## 1 Introduction

Deep learning has rapidly become one of the most pervasively applied techniques in machine learning. From computer vision (Krizhevsky et al. (2012)) and reinforcement learning (Mnih et al. (2013)) to natural language processing (Wu et al. (2016)) and speech recognition (Hinton et al. (2012)), the core principles of hierarchical representation and optimization central to deep learning have revolutionized the state of the art; see Goodfellow et al. (2016). In each domain, a major difficulty lies in selecting the architectures of models that most optimally take advantage of structure in the data. In computer vision, for example, a large body of work (Simonyan & Zisserman (2014), Szegedy et al. (2014), He et al. (2015), etc.) focuses on improving the initial architectural choices of Krizhevsky et al. (2012) by developing novel network topologies and optimization schemes specific to vision tasks. Despite the success of this approach, there are still not general principles for choosing architectures in arbitrary settings, and in order for deep learning to scale efficiently to new problems and domains without expert architecture designers, the problem of architecture selection must be better understood.

Theoretically, substantial analysis has explored how various properties of neural networks, (eg. the depth, width, and connectivity) relate to their expressivity and generalization capability (Raghu et al. (2016), Daniely et al. (2016), Guss (2016)). However, the foregoing theory can only be used to determine an architecture in practice if it is understood how expressive a model need be in order to solve a problem. On the other hand, neural architecture search (NAS) views architecture selection as a compositional hyperparameter search (Saxena & Verbeek (2016), Fernando et al. (2017), Zoph & Le (2017)). As a result NAS ideally yields expressive and powerful architectures, but it is often difficult to interperate the resulting architectures beyond justifying their use from their emperical optimality.

We propose a third alternative to the foregoing: data-first architecture selection. In practice, experts design architectures with some inductive bias about the data, and more generally, like any hyperparameter selection problem, the most expressive neural architectures for learning on a particular dataset are solely determined by the nature of the true data distribution. Therefore, architecture selection can be rephrased as follows: *given a learning problem (some dataset), which architectures are suitably regularized and expressive enough to learn and generalize on that problem?*

A natural approach to this question is to develop some objective measure of data complexity, and then characterize neural architectures by their ability to learn subject to that complexity. Then given

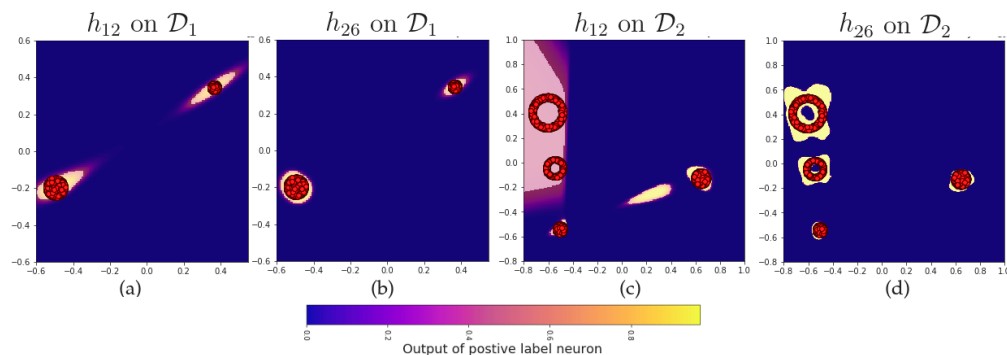

Figure 1: The positive label outptus of single hidden layer neural networks, $h_{12}$ and $h_{26}$, of 2 inputs with 12 and 26 hidden units respectively after training on datasets $\mathcal{D}_1$ and $\mathcal{D}_2$ with positive examples in red. Highlighted regions of the output constitute the positive decision region.

some new dataset, the problem of architecture selection is distilled to computing the data complexity and chosing the appropriate architecture.

For example, take the two datasets $\mathcal{D}_1$ and $\mathcal{D}_2$ given in Figure 1(ab) and Figure 1(cd) respectively. The first dataset, $\mathcal{D}_1$, consists of positive examples sampled from two disks and negative examples from their compliment. On the right, dataset $\mathcal{D}_2$ consists of positive points sampled from two disks and two rings with hollow centers. Under some geometric measure of complexity $\mathcal{D}_2$ appears more 'complicated' than $\mathcal{D}_1$ because it contains more holes and clusters. As one trains single layer neural networks of increasing hidden dimension on both datasets, *the minimum number of hidden units required to achieve zero testing error is ordered according to this geometric complexity.* Visually in Figure 1, regardless of initialization no single hidden layer neural network with $\leq 12$ units, denoted $h_{\leq 12}$, can express the two holes and clusters in $\mathcal{D}_2$. Whereas on the simpler $\mathcal{D}_1$, both $h_{12}$ and $h_{26}$ can express the decision boundary perfectly. Returning to architecture selection, one wonders if this characterization can be extrapolated; that is, is it true that for datasets with 'similar' geometric complexity to $\mathcal{D}_1$, any architecture with $\geq 12$ hidden learns perfectly, and likewise for those datasets similar in complexity to $\mathcal{D}_2$, architectures with $\leq 12$ hidden units can never learn to completion?

## 1.1 OUR CONTRIBUTION

In this paper, we formalize the above of geometric complexity in the language of algebraic topology. We show that questions of architecture selection can be answered by understanding the 'topological capacity' of different neural networks. In particular, a geometric complexity measure, called persistent homology, characterizes the capacity of neural architectures in direct relation to their ability to generalize on data. Using persistent homology, we develop a method which gives the first empirical insight into the learnability of different architectures as data complexity increases. In addition, our method allows us to generate conjectures which tighten known theoretical bounds on the expressivity of neural networks. Finally, we show that topological characterizations of architectures are possible and useful for architecture selection in practice by computing the persistent homology of CIFAR-10 and several UCI datasets.

## 2 BACKGROUND

### 2.1 GENERAL TOPOLOGY

In order to more formally describe notions of geometric complexity in datasets, we will turn to the language of topology. Broadly speaking, topology is a branch of mathematics that deals with characterizing shapes, spaces, and sets by their *connectivity*. In the context of characterizing neural networks, we will work towards defining the topological complexity of a dataset in terms of how that dataset is 'connected', and then group neural networks by their capacity to produce decision regions of the same connectivity.

In topology, one understands the relationships between two different spaces of points by the *continuous maps* between them. Informally, we say that two topological spaces $A$ and $B$ are *equivalent* ($A \cong B$) if there is a continuous function $f : A \to B$ that has an inverse $f^{-1}$ that is also continuous. When $f$ exists, we say that $A$ and $B$ are *homeomorphic* and $f$ is their *homeomorphism*; for a more detailed treatment of general topology see Bredon (2013). Take for example, the classic example of the coffee cup and the donut in

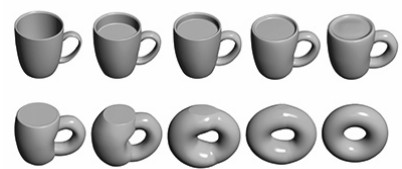

Figure 2: A continuous deformation of a coffee cup into a donut, showing that both are topologically equivalent (Kato et al. (2014)).

Figure 2. They are homeomorphic because one can define a continuous deformation of one into the other which shrinks, twists, and morphs without tearing or gluing, as in Figure 2. Note that if the donut had two holes, it would no longer be equivalent to the mug. Likewise, in an informal way, $\mathcal{D}_1 \not\cong \mathcal{D}_2$ in Figure 1 since if there were a homeomorphism $f : \mathcal{D}_1 \to \mathcal{D}_2$ at least one of the clusters in $\mathcal{D}_1$ would need to be split in order to produce the four different regions in $\mathcal{D}_2$.

The power of topology lies in its capacity to differentiate sets (topological spaces) in a meaningful geometric way that discards certain irrelevant properties such as rotation, translation, curvature, etc. For the purposes of defining geometric complexity, non-topological properties[1] like curvature would further fine-tune architecture selection–say if $\mathcal{D}_2$ had the same regions but with squigly (differentially complex) boundaries, certain architectures might not converge–but as we will show, grouping neural networks by 'topological capacity' provides a powerful minimality condition. That is, we will show that if a certain architecture is incapable of expressing a decision region that is equivalent in topology to training data, then there is no hope of it ever generalizing to the true data.

## 2.2 ALGEBRAIC TOPOLOGY

Algebraic topology provides the tools necessary to not only build the foregoing notion of topological equivalence into a measure of geometric complexity, but also to compute that measure on real data (Betti (1872), Dey et al. (1998), Bredon (2013)). At its core, algebraic topology takes topological spaces (shapes and sets with certain properties) and assigns them algebraic objects such as *groups*, *chains*, and other more exotic constructs. In doing so, two spaces can be shown to be topologically equivalent (or distinct) if the algebraic objects to which they are assigned are isomorphic (or not). Thus algebraic topology will allow us to compare the complexity of decision boundaries and datasets by the objects to which they are assigned.

Although there are many flavors of algebraic topology, a powerful and computationally realizable tool is homology.

**Definition 2.1** (Informal, Bredon (2013)). *If $X$ is a topological space, then $H_n(X) = \mathbb{Z}^{\beta_n}$ is called the $n$th homology group of $X$ if the power $\beta_n$ is the number of 'holes' of dimension $n$ in $X$. Note that $\beta_0$ is the number of separate connected components. We call $\beta_n(X)$ the $n$th Betti number of $X$. Finally, the homology[2] of $X$ is defined as $H(X) = \{H_n(X)\}_{n=0}^{\infty}$.*

Immediately homology brings us closer to defining the complexity of $\mathcal{D}_1$ and $\mathcal{D}_2$. If we assume that $\mathcal{D}_1$ is not actually a collection of $N$ datapoints, but really the union of 2 solid balls, and likewise that $\mathcal{D}_2$ is the union of 2 solid balls and 2 rings, then we can compute the homology directly. In this case $H_0(\mathcal{D}_1) = \mathbb{Z}^2$ since there are two connected components[3]; $H_1(\mathcal{D}_1) = \{0\}$ since there are no circles (one-dimensional holes); and clearly, $H_n(\mathcal{D}_1) = \{0\}$ for $n \geq 2$. Performing the same computation in the second case, we get $H_0(\mathcal{D}_2) = \mathbb{Z}^4$ and $H_1(\mathcal{D}_2) = \mathbb{Z}^2$ as there are 4 seperate clusters and 2 rings/holes. With respect to any reasonable ordering on homology, $\mathcal{D}_2$ is more complex than $\mathcal{D}_1$. The measure yields non-trivial differentiation of spaces in higher dimension. For example, the homology of a hollow donut is $\{\mathbb{Z}^1, \mathbb{Z}^2, \mathbb{Z}^1, 0, \dots\}$.

---

[1]A *topological property* or *invariant* is one that is preserved by a homeomorphism. For example, the number of holes and regions which are disjoint from one another are topological properties, whereas curvature is not.

[2]This definition of homology makes many assumptions on $X$ and the base field of computation, but for introductory purposes, this informality is edifying.

[3]Informally, a *connected component* is a set which is not contained in another connected set except for itself.

Surprisingly, the homology of a space contains a great deal of information about its topological complexity[1]. The following theorem suggests the absolute power of homology to group topologically similar spaces, and therefore neural networks with topologically similar decision regions.

**Theorem 2.2** (Informal). *Let $X$ and $Y$ be topological spaces. If $X \cong Y$ then $H(X) = H(Y)$.*[4]

Intuitively, Theorem 2.2 states that number of 'holes' (and in the case of $H_0(X)$, connected components) are topologically invariant, and can be used to show that two shapes (or decision regions) are different.

## 2.3 Computational Methods for Homological Complexity

In order to compute the homology of both $\mathcal{D}_1$ and $\mathcal{D}_2$ we needed to assume that they were actually the geometric shapes from which they were sampled. Without such assumptions, *for any dataset* $\mathcal{D}$ a $H(\mathcal{D}) = \{\mathbb{Z}^N, 0, \dots\}$ where $N$ is the number of data points. This is because, at small enough scales each data point can be isolated as its own connected component; that is, as sets each pair of different positive points $d_1, d_2 \in \mathcal{D}$ are disjoint. To properly utilize homological complexity in better understanding architecture selection, we need to be able to compute the homology of the data directly and still capture meaningful topological information.

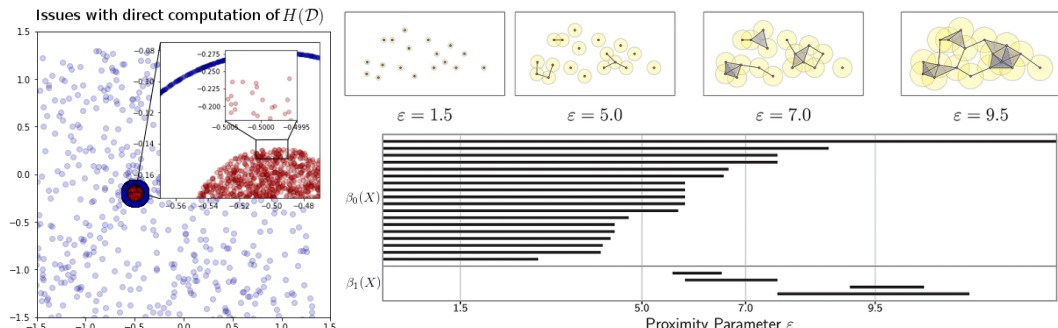

Figure 3: Left: The local disconnectedness of datasets prevents direct computation of their homology. Right: An illustration of computing persistent homology on a collection of points (Topaz et al. (2015))

Persistent homology, introduced in Zomorodian & Carlsson (2005), avoids the trivialization of computation of dataset homology by providing an algorithm to calculate the homology of a *filtration* of a space. Specifically, a filtration is a topological space $X$ equipped with a sequence of subspaces $X_0 \subset X_1 \subset \cdots \subset X$. In Figure 3 one such particular filtration is given by growing balls of size $\epsilon$ centered at each point, and then letting $X_\epsilon$ be the resulting subspace in the filtration. Define $\beta_n(X)$ to be the $n$th Betti number of the homology $H(X_\epsilon)$ of $X_\epsilon$. Then for example at $\epsilon = 1.5$, $\beta_0(X_\epsilon) = 19$ and $\beta_1(X_\epsilon) = 0$ as every ball is disjoint. At $\epsilon = 5.0$ some connected components merge and $\beta_0(X_\epsilon) = 12$ and $\beta_1(X_\epsilon) = 0$. Finally at $\epsilon = 7$, the union of the balls forms a hole towards the center of the dataset and $\beta_1(X_\epsilon) > 0$ with $\beta_0(X_\epsilon) = 4$.

All together the change in homology and therefore Betti numbers for $X_\epsilon$ as $\epsilon$ changes can be summarized succinctly in the *persistence barcode diagram* given in Figure 3. Each bar in the section $\beta_n(X)$ denotes a 'hole' of dimension $n$. The left endpoint of the bar is the point at which homology detects that particular component, and the right endpoint is when that component becomes indistinguishable in the filtration. When calculating the persistent homology of datasets we will frequently use these diagrams.

With the foregoing algorithms established, we are now equipped with the tools to study the capacity of neural networks in the language of algebraic topology.

---

[4]Equality of $H(X)$ and $H(Y)$ should be interpreted as isomorphism between each individual $H_i(X)$ and $H_i(Y)$.

## 3 Homological Characterization of Neural Architectures

In the forthcoming section, we will apply persistent homology to emperically characterize the power of certain neural architectures. To understand why homological complexity is a powerful measure for differentiating architectures, we present the following principle.

Suppose that $\mathcal{D}$ is some dataset drawn from a joint distribution $F$ with continuous CDF on some topological space $X \times \{0, 1\}$. Let $X^+$ denote the support of the distribution of points with positive labels, and $X^-$ denote that of the points with negative labels. Then let $H_S(f) := H[f^{-1}((0, \infty))]$ denote the *support homology* of some function $f : X \to \{0, 1\}$. Essentially $H_S(f)$ is homology of the set of $x$ such that $f(x) > 0$. For a binary classifier, $f$, $H_S(f)$ is roughly a characterization of how many 'holes' are in the positive decision region of $f$. We will sometimes use $\beta_n(f)$ to denote the $n$th Betti number of this support homology. Finally let $\mathcal{F} = \{f : X \to \{0, 1\}\}$ be some family of binary classifiers on $X$.

**Theorem 3.1** (The Homological Principle of Generalization). *If $X = X^- \sqcup X^+$ and for all $f \in \mathcal{F}$ with $H_S(f) \neq H(X^+)$, then for all $f \in \mathcal{F}$ there exists $A \subset X^+$ so $f$ misclassifies every $x \in A$.*

Essential Theorem 3.1 says that if an architecture (a family of models $\mathcal{F}$) is incapable of producing a certain homological complexity, then for any model using that architecture there will always be a set $A$ of true data points on which the model will fail. Note that the above principle holds regardless of how $f \in \mathcal{F}$ is attained, learned or otherwise. However, the principle does imply that no matter how well some $\mathcal{F}$ learns to correctly classify $\mathcal{D}$ there will always be a counter examples in the true data.

In the context of architecture selection, the foregoing minimality condition significantly reduces the size of the search space by eliminating smaller architectures which cannot even express the 'holes' (persistent homology) of the data $H(\mathcal{D})$. This allows us to return to our original question of finding suitably expressive and generalizeable architectures but in the very computable language of homological complexity: Let $\mathcal{F}_A$ the set of all neural networks with 'architecture' $A$, then

*Given a dataset $\mathcal{D}$, for which architectures $A$ does there exist*
*a neural network $f \in \mathcal{F}_A$ such that $H_S(f) = H(\mathcal{D})$?*

We will resurface a contemporary theoretical view on this question, and thereafter make the first steps towards an emperical characterization of the capacity of neural architectures in the view of topology.

### 3.1 Theoretical Basis for Neural Homology

Theoretically, the homological complexity of neural network can be framed in terms of the sum of the number of holes expressible by certain architectures. In particular, Bianchini et al. (2014) gives an analysis of how the maximum sum of Betti numbers grows as $\mathcal{F}_A$ changes. The results, summarized in Table 1, show that the width, depth, and activation of a fully connected architecture effect its topological expressivity to varying polynomial and exponential degrees. What is unclear from this analysis is how these bounds describe expressivity in terms of individual Betti numbers. For example, with a $\tanh$ activation function, $n$ inputs, $\ell$ layers, and $h$ hidden units, there is no description of what the number of connected components $\max_{f \in \mathcal{F}_A} \beta_0(f)$ or 1-dimensional holes $\max_{f \in \mathcal{F}_A} \beta_1(f)$ actually is. With regards to tighter bounds Bianchini et al. (2014) stipulate that improvements to their results are deeply tied to several unsolved problems in algebraic topology.

Table 1: Upper bounds on homological expressivity of neural architectures (Bianchini et al. (2014).)

| Architecture $A$ | | | | $\max_{f \in \mathcal{F}_A} \sum_{k=1}^{n} \beta_k(f)$ |
|---|---|---|---|---|
| Inputs | Layers | Units | Activation | |
| $n$ | 3 | h | threshold | $O(h^n)$ |
| $n$ | 3 | h | arctan | $O((n+h)^{n+2})$ |
| $n$ | 3 | h | polynomial, deg. $r$ | $\frac{1}{2}(2+r)(1+r)^{n-1}$ |
| 1 | 3 | h | arctan | $h$ |
| $n$ | $\ell$ | h | arctan | $2^{h(2h-1)}O((n\ell+n)^{n+2h})$ |
| $n$ | $\ell$ | h | tanh | $2^{h(h-1)/2}O((n\ell+n)^{n+h})$ |
| $n$ | $\ell$ | h | polynomial, deg. $r$ | $\frac{1}{2}(2+r)(1+r)^{n-1}$ |

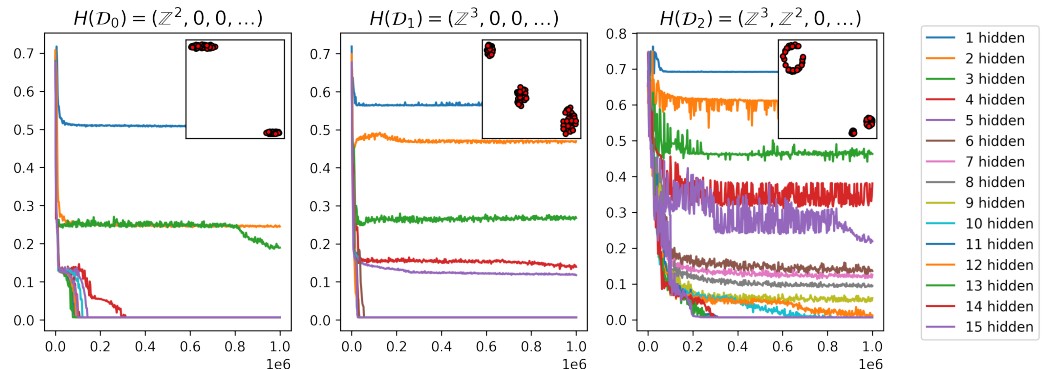

Figure 4: Topological phase transitions in low dimensional neural networks as the homological complexity of the data increases. The upper right corner of each plot is a dataset on which the neural networks of increasing hidden dimension are trained.

## 3.2 EMPIRICAL RESULTS

To understand how the homology of data determines expressive architectures we turn to an empirical characterization of neural networks. In this setting, we can tighten the bounds given in Table 1 by training different architectures on datasets with known homologies and then recording the decision regions observed over the course of training.

### 3.2.1 HOMOLOGICAL CAPACITY OF HIDDEN UNITS.

In the most basic case, one is interested in studying how the number of hidden units in a single hidden layer neural network affect its homological capacity. The results of Bianchini et al. (2014) say for certain activation functions we should expect a polynomial dependence on the sum of Betti numbers $\sum \beta_n$, but is this true of individual numbers? Having an individual characterization would allow for architecture selection by computing the homology of the dataset, and then finding which architectures meet the minimal criterion for each Betti number $\beta_n$.

Restricting[5] our analysis to the case of two inputs, $n = 2$, we characterize the capacities of architectures with an increasing number of hidden units to learn on datasets with homological complexities ranging from $\{\mathbb{Z}^1, 0\}$ to $\{\mathbb{Z}^{20}, \mathbb{Z}^{20}\}$. In our experiment, we generate datasets of each particular homological complexity by sampling different combinations of balls, rings, and scaling, twisting, and gluing them at random. After generating the foregoing datasets with $N \approx 90000$ samples we train 100 randomly (truncated normal) initialized single hidden layer architectures with hidden units $h \in \{1, \ldots, 255\}$ and $\tanh$ activation functions for $10^6$ minibatches of size 128. During training, every 2500 batches we sample the decision boundary of each neural network over a grid of $500 \times 500$ samples, producing $1.02 \times 10^6$ recorded decision boundaries. Using the resulting data, we not only characterize different architectures but observed interesting topological phenomena during learning.

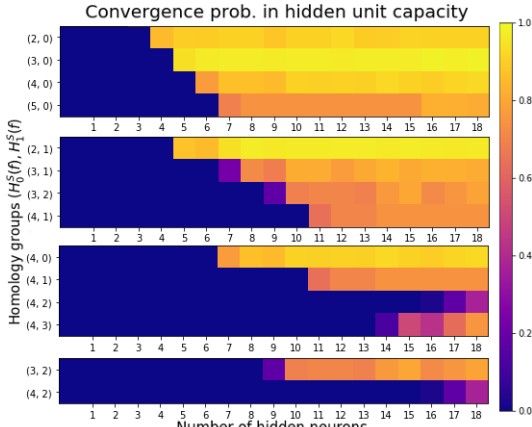

Figure 5: Several different views of the probability of converging to zero-error for single hidden layer neural networks on datasets with different homological complexities.

---

[5]Analysis of greater input dimension was not given due to space constraints (decision boundary samples alone accounted for 18.55 TB), but in future work Monte Carlo samples of the decision boundary suffice to perform the forthcoming analysis; see Chazal et al. (2015).

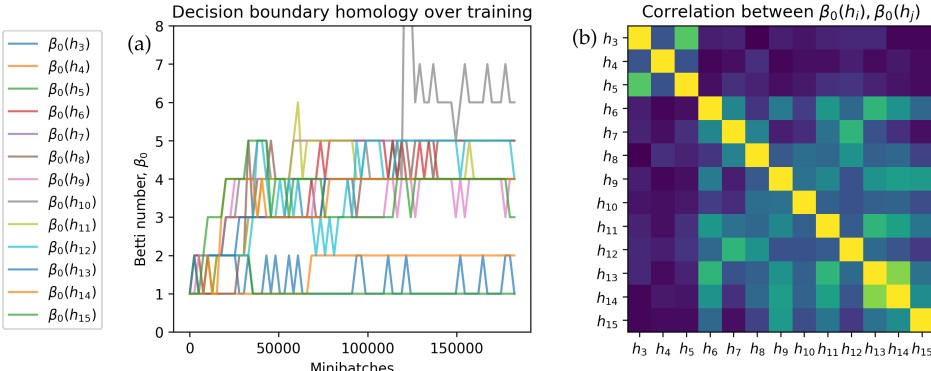

Figure 6: An example of topological stratification for single hidden layer networks. (a) The number of connected components in the decision regions during training. (b) Correlation of Betti numbers.

First, neural networks exhibit a statistically significant *topological phase transition* in their convergence which depends directly on the homological complexity of the data. For any dataset in the experiment and any random homeomorphism applied thereto, the best test error of architectures with $h$ hidden units is *strictly* ordered in magnitude and convergence time for $h < h_{phase}$ where $h_{phase}$ is a number of hidden units required to express the homology of the data. In Figure 4 we plot the best performing test error of architectures $h \in \{1, \ldots, 15\}$ on some example datasets $\mathcal{D}_0, \mathcal{D}_1$, and $\mathcal{D}_2$ with $H(\mathcal{D}_0) \approx \{\mathbb{Z}^2, 0\}$, $H(\mathcal{D}_1) \approx \{\mathbb{Z}^3, 0\}$, $H(\mathcal{D}_2) \approx \{\mathbb{Z}^3, \mathbb{Z}^2\}$. In this example $h_{phase}(\mathcal{D}_0) = 4, h_{phase}(\mathcal{D}_1) = 6$, and $h_{phase}(\mathcal{D}_2) = 10$. Surprisingly, leading up to the phase transition point, each different architecture falls into its own band of optimal convergence. This suggests that additional hidden units do in fact add to the topological capacity of an architecture in a consistent way.

Using topological phase transitions we now return to the original question of existence of expressive architectures. In Figure 5, we accumulate the probabilities that neural networks of varying hidden dimension train to zero-error on datasets of different homological complexities. The table gives different views into how expressive an architecture need be in order to converge, and therefore we are able to conjecture tighter bounds on the capacity of hidden units. Extrapolating from the first view, if $H_0(\mathcal{D}) = \mathbb{Z}^m$ then there exists a single hidden layer neural network with $h = m + 2$ that converges to zero error on $\mathcal{D}$. Likewise we claim that if $H_0(\mathcal{D}) = \mathbb{Z}^m$ and $H_1(\mathcal{D}) = 1$ then the same holds with $h \geq 3m - 1$. Further empirical analysis of convergence probabilities yields additional conjectures. However, claiming converse conjectures about a failure to generalize in the view of Theorem 3.1 requires exhaustive computation of decision boundary homologies.

By applying persistent homology to the decision boundaries of certain networks during training, we observe that given sufficient data, neural networks exhibit *topological stratification*. For example, consider the homologies of different architecture decision regions as training progresses in Figure 6(a). At the beginning of training every model captures the global topological information of the dataset and is homologically correlated with one another. However as training continues, the architectures stratify into two groups with homological complexities ordered by the capacities of the models. In this example, $h_3, h_4$, and $h_5$ are unable to express as many holes as the other architectures and so never specialize to more complex and local topological properties of the data. Figure 6(b) depicts topological stratification in terms of the correlation between Betti numbers. Topologically speaking, networks with less than 6 hidden units are distinct from those with more for most of training. Furthermore, this correlative view shows that stratification is consistent with topological phase transition; that is, across all decision boundary homologies recorded during the experiment stratification occurs just when the number of hidden units is slightly less than $h_{phase}$.

## 4 THE TOPOLOGY OF REAL DATA

We have thus far demonstrated the discriminatory power of homological complexity in determining the expressivity of architectures. However, for homological complexity to have any practical use in architecture selection, it must be computable on real data, and more generally real data must have non-trivial homology; if all data were topologically simple our characterization would have no

Figure 7: The persistent homology barcodes of classes in the CIFAR-10 and UCI Protein Localization Datasets. Left: The bardcode for dimension $0$ and $1$ of the 'CYT' class along side its local linear embedding into $\mathbb{R}^2$. Right: The barcode for the dimensions $0$ and $1$ for the 'cars' class along side different samples thereof in CIFAR-10. Note how different orientations are shown.

predictive power. In the following section we will compute the persistent homologies up to dimension 2 of different real world datasets.

**CIFAR-10.** We compute the persistent homology of several classes of CIFAR-10 using the Python library Dionysus. Currently algorithms for persistent homology do not deal well with high dimensional data, so we embed the entire dataset in $\mathbb{R}^3$ using local linear embedding (LLE; Saul & Roweis (2000)) with $K = 120$ neighbors. After embedding the dataset, we take a sample of $1000$ points from example class 'car' and build a persistent filtration by constructing a Vietoris-Rips complex on the data. The resulting complex has $20833750$ simplices and took $4.3$ min. to generate. Finally, computation of the persistence diagram shown in Figure 7 took $8.4$ min. locked to a single thread on a Intel Core i7 processor. The one-time cost of computing persistent homology could easily augment any neural architecture search.

Although we only give an analysis of dimension 2 topological features–and there is certainly higher dimensional homological information in CIFAR-10–the persistence barcode diagram is rich with different components in both $H_0(\mathcal{D})$ and $H_1(\mathcal{D})$. Intuitively, CIFAR contains pictures of cars rotated accross a range of different orientations and this is exhibited in the homology. In particular, several holes are born and die in the range $\epsilon \in [0.15, 0.375]$ and one large loop from $\epsilon \in [0.625, 0.82]$.

**UCI Datasets.** We further compute the homology of three low dimensional UCI datasets and attempt to assert the of non-trivial , $h_{phase}$. Specifically, we compute the persistent homology of the majority classes in the Yeast Protein Localization Sites, UCI Ecoli Protein Localization Sites, and HTRU2 datasets. For these datasets no dimensionality reduction was used. In Figure 7(left), the persistence barcode exhibits two seperate significant loops (holes) at $\epsilon \in [0.19, 0.31]$ and $\epsilon \in [0.76, 0.85]$, as well as two major connected components in $\beta_0(\mathcal{D})$. The Other persistence diagrams are relegated to the appendix.

**Existing Data**. Outside of the primary machine learning literature, topological data analysis yields non-trivial computations in wide variety of fields and datasets. Of particular interest is the work of Carlsson et al. (2008), which computes the homological complexity of collections of $n \times n$ patches of natural images. Even in these simple collections of images, the authors found topologies of Klein Bottles ($H(\cdot) = \{\mathbb{Z}, \mathbb{Z}^2/2\mathbb{Z}, 0 \dots\}$) and other exotic topological objects. Other authors have calculated non-trivial dataset homologies in biological (Topaz et al. (2015)), natural language (Michel et al. (2017)), and other domains (Wu et al. (2017), Xia & Wei (2014)).

## 5 RELATED WORK

We will place this work in the context of deep learning theory as it relates to expressivity. Since the seminal work of Cybenko (1989) which established standard universal approximation results for neural networks, many theorists have endeavored to understand the expressivity of certain neural architectures. Pascanu et al. (2013) and MacKay (2003) provided the first analysis relating the depth and width of architectures to the complexity of the sublevel sets they can express. Motivated therefrom, Bianchini et al. (2014) expressed this theme in the language of Pfefferian functions, thereby bounding the sum of Betti numbers expressed by sublevel sets. Finally Guss (2016) gave an account of how topological assumptions on the input data lead to optimally expressive architectures. In parallel, Eldan & Shamir (2016) presented the first analytical minimality result in expressivity theory; that is, the authors show that there are simple functions that cannot be expressed by two layer

neural networks with out exponential dependence on input dimension. This work spurred the work ofPoole et al. (2016), Raghu et al. (2016) which reframed expressivity in a differential geometric lense.

Our work presents the first method to derive expressivity results empirically. Our topological viewpoint sits dually with its differential geometric counterpart, and in conjunction with the work of Poole et al. (2016) and Bianchini et al. (2014), this duallity implies that when topological expression is not possible, exponential differential expressivity allows networks to bypass homological constaints at the cost of adversarial sets. Furthermore, our work opens a practical connectio nbetween the foregoing theory on neural expressivity and architecture selection, with the potential to drastically improve neural architecture search (Zoph & Le (2017)) by directly computing the capacities of different architectures.

# 6 CONCLUSION

Architectural power is deeply related to the algebraic topology of decision boundaries. In this work we distilled neural network expressivity into an empirical question of the generalization capabilities of architectures with respect to the homological complexity of learning problems. This view allowed us to provide an empirical method for developing tighter characterizations on the the capacity of different architectures in addition to a principled approach to guiding architecture selection by computation of persistent homology on real data.

There are several potential avenues of future research in using homological complexity to better understand neural architectures. First, a full characterization of neural networks with many layers or convolutional linearities is a crucial next step. Our empirical results suggest that the their are exact formulas describing the of power of neural networks to express decision boundaries with certain properties. Future theoretical work in determining these forms would significantly increase the efficiency and power of neural architecture search, constraining the search space by the persistent homology of the data. Additionally, we intend on studying how the topological complexity of data changes as it is propagated through deeper architectures.

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

## A PROOFS, CONJECTURES, AND FORMAL DEFINITIONS

### A.1 HOMOLOGY

Homology is naturally described using the language of category theory. Let $Top^2$ denote the category of topological spaces and $Ab$ the category of abelian groups.

**Definition A.1** (Homology Theory, Bredon (2013)). *A homology theory on the on $Top^2$ is a function $H : Top^2 \to Ab$ assigning to each pair $(X, A)$ of spaces a graded (abelian) group $\{H_p(X, A)\}$, and to each map $f : (X, A) \to (Y, B)$, homomorphisms $f_* : H_p(X, A) \to H_p(Y, B)$, together with a natural transformation of functors $\partial_* : H_p(X, A) \to H_{p-1}(X, A)$, called the connecting homomorphism (where we use $H_*(A)$ to denote $H_*(A, \emptyset)$) such that the following five axioms are satisfied.*

1. *If $f \simeq g : (X, A) \to (Y, B)$ then $f_* = g_* : H_*(X, A) \to H_*(Y, B)$.*

2. *For the inclusions $i : A \to X$ and $j : X \to (X, A)$ the sequence sequence of inclusions and connecting homomorphisms are exact.*

3. *Given the pair $(X, A)$ and an open set $U \subset X$ such that $cl(U) \subset int(A)$ then the inclusion $k : (X - U, A - U) \to (X, A)$ induces an isomorphism $k_* : H_*(X - U, A - U) \to H_*(X, A)$*

4. *For a one point space $P$, $H_i(P) = 0$ for all $i \neq 0$.*

5. *For a topological sum $X = +_\alpha X_\alpha$ the homomorphism*

$$\bigoplus (i_\alpha)_* : \bigoplus H_n(X_\alpha) \to H_n(X)$$

*is an isomorphism, where $i_\alpha : X_\alpha \to X$ is the inclusion.*

For related definitions and requisite notions we refer the reader to Bredon (2013).

### A.2 PROOF OF THEOREM 3.1

**Theorem A.2.** *Let $X$ be a topological space and $X^+$ be some open subspace. If $\mathcal{F} \subset 2^X$ such that $f \in \mathcal{F}$ implies $H_S(f) \neq H(X^+)$, then for all $f \in \mathcal{F}$ there exists $A \subset X$ so that $f(A \cap X^+) = \{0\}$ and $f(A \cap (X \setminus X^+)) = \{1\}$.*

*Proof.* Suppose the for the sake of contraiction that for all $f \in \mathcal{F}$, $H_S(f) \neq H(X^+)$ and yet there exists an $f$ such that for all $A \subset X$, there exists an $x \in A$ such that $f(x) = 1$. Then take $\mathcal{A} = \{x\}_{x \in X}$, and note that $f$ maps each singleton into its proper partition on $X$. We have that for any open subset of $V \subset X^+$, $f(V) = \{1\}$, and for any closed subset $W \subset X \setminus X^+$, $f(W) = \{0\}$. Therefore $X^+ = \bigcup_{A \in \tau_{X^+ \cap X}} A \subset supp(f)$ as the subspace topology $\tau_{X^+ \cap X} = \tau_{X^+} \cap \tau_X$ where $\tau_{X^+} = \{A \in \tau_X \mid A \subset X^+\}$ and $\tau_X$ denotes the topology of $X$. Likewise, $int(X^-) \subset X \setminus supp(F)$ under the same logic. Therefore $supp(f)$ has the exact same topology as $X^+$ and so by Theorem 2.2 $H(X^+) = H(supp(f))$ but this is a contradiction. This completes the proof. $\square$

### A.3 THE NEURAL HOMOLOGY PRINCIPLE

**Conjecture A.3.** *If $N$ is some neural network with $\ell$ layers and $h$ hidden units, and $H_S(N) \neq H(X^+)$ then $\mathbb{E}[L(N, \mathcal{D})] > c$ for some fixed $c(\ell, h) > 0$.*

# B   ADDITIONAL TOPOLOGICAL PHASE TRANSITIONS

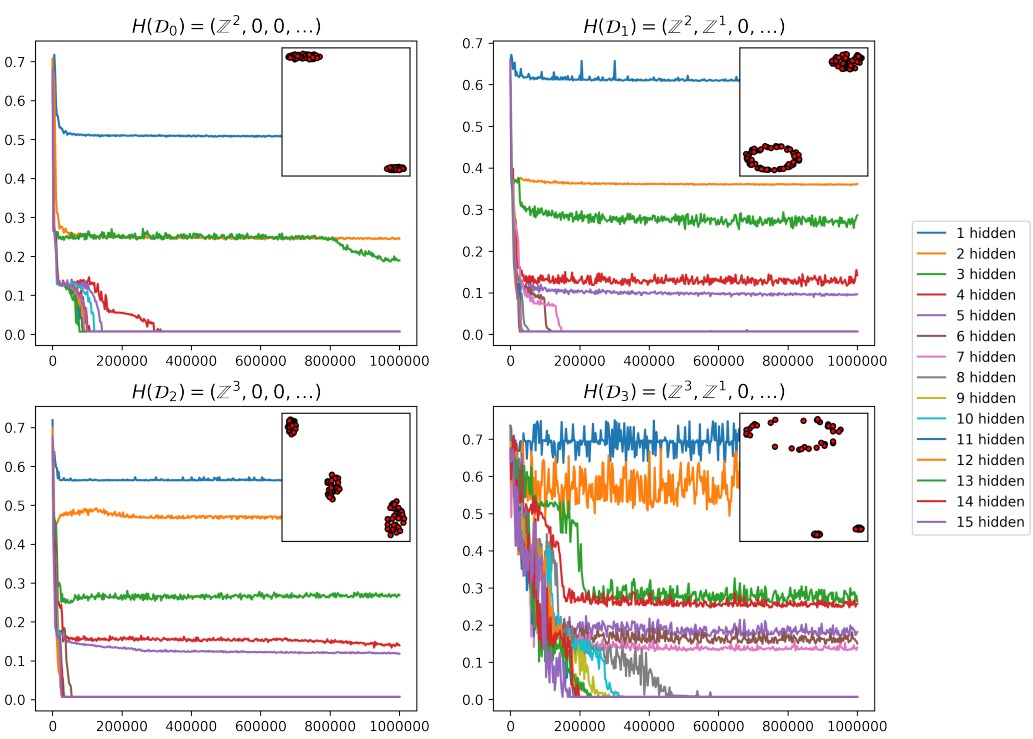

Figure 8: Topological phase transitions for datasets with $\beta(\mathcal{D}) \in \{(2,0),(2,1),(3,0),(3,1)\}$

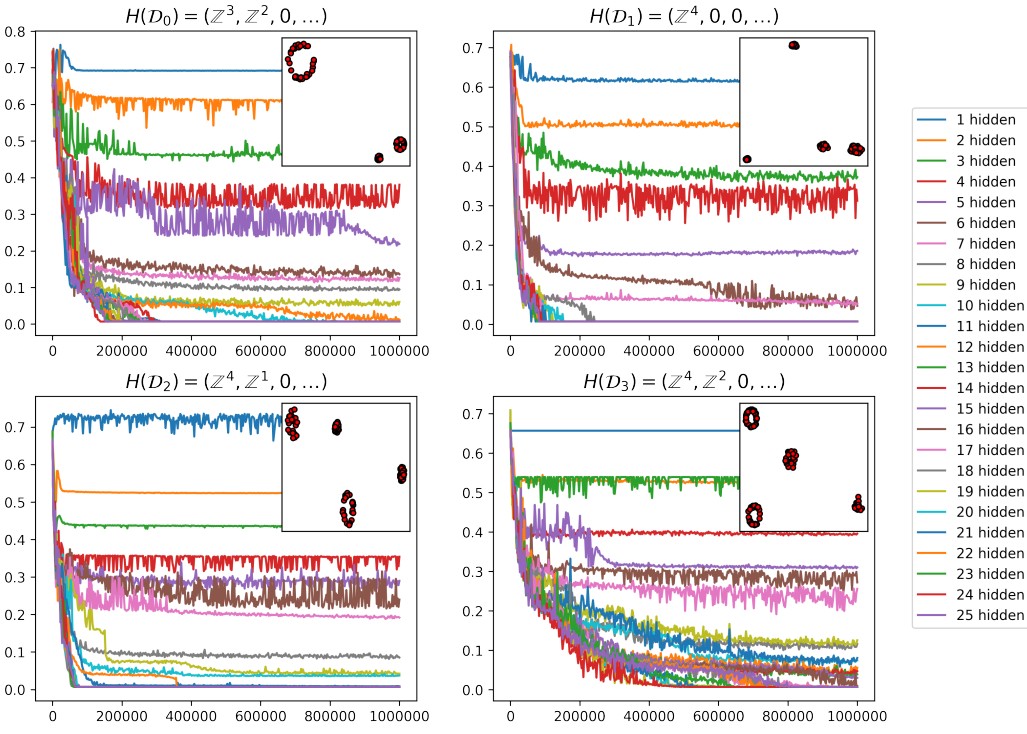

Figure 9: Topological phase transitions for datasets with $\beta(\mathcal{D}) \in \{(3,2),(4,0),(4,1),(4,2)\}$

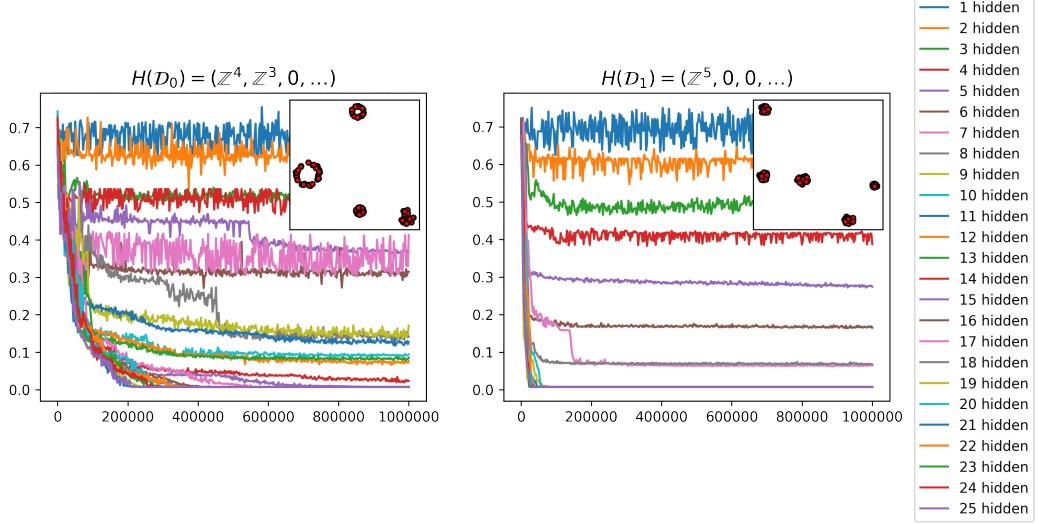

Figure 10: Topological phase transitions for datasets with $\beta(\mathcal{D}) \in \{(4,3),(5,0)\}$

## C EXAMPLE TOPOLOGICAL STRATIFICATIONS

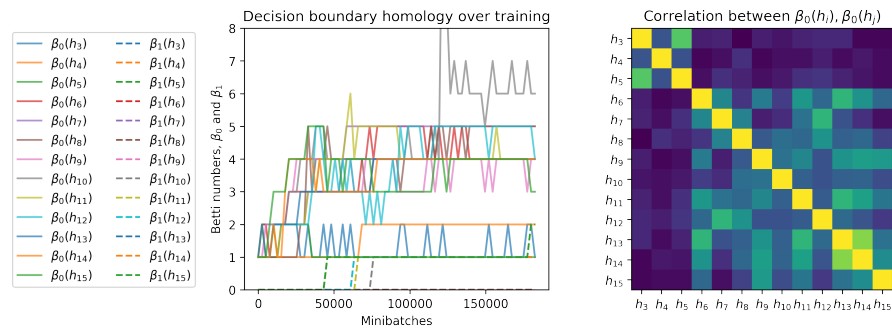

Figure 11: Topological stratification for $\{h_3, \ldots, h_{15}\}$ on a random dataset $\mathcal{D}$ with $\beta_0(\mathcal{D}) = 25$, $\beta_1(\mathcal{D}) = 16$.

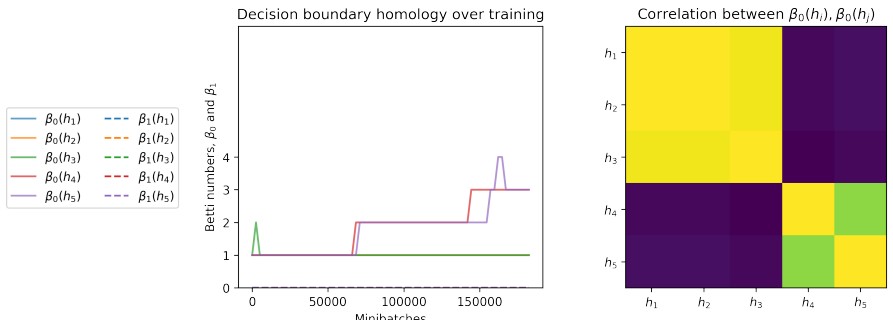

Figure 12: Topological stratification for $\{h_1, \ldots, h_5\}$ on a random dataset $\mathcal{D}$ with $\beta_0(\mathcal{D}) = 3$, $\beta_1(\mathcal{D}) = 0$.

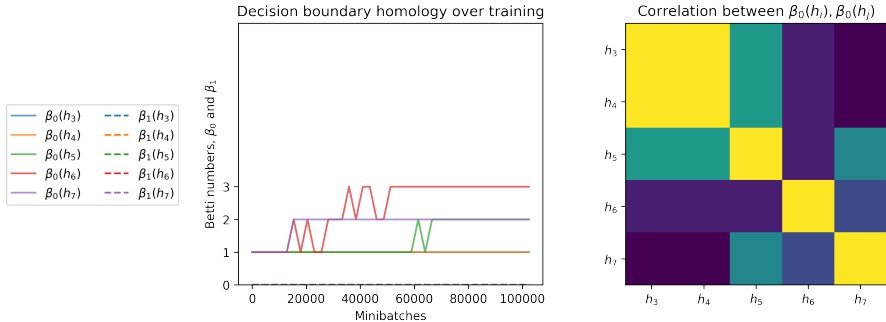

Figure 13: Topological stratification for $\{h_3, \ldots, h_7\}$ on a random dataset $\mathcal{D}$ with $\beta_0(\mathcal{D}) = 3$, $\beta_1(\mathcal{D}) = 0$.

# D    ADDITIONAL TOPOLOGY OF REAL DATA

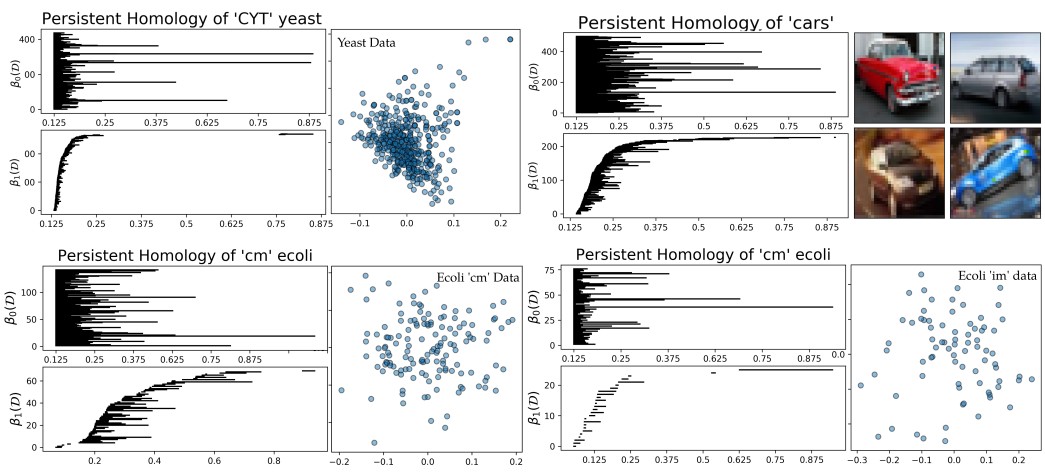

Figure 14: The persistence diagrams of other data.

# E    EXAMPLE SAMPLED DATASETS

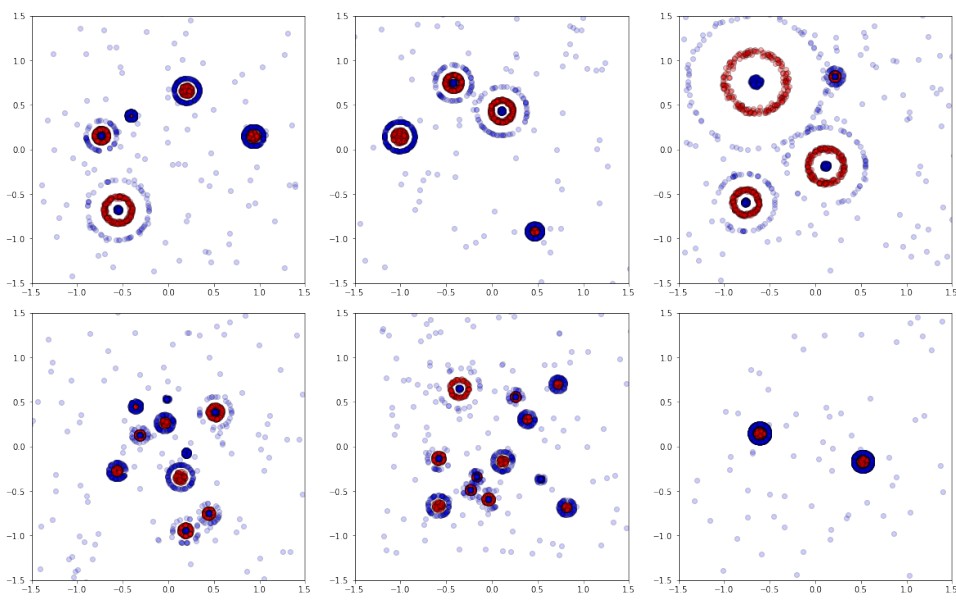

