# OpenReview forum: "On Characterizing the Capacity of Neural Networks Using Algebraic Topology"
_ICLR.cc/2018/Conference — Reject_

### Official Review · AnonReviewer2 · 2017-11-27

**Rating:** 3
**Confidence:** 5

**Review:**

Paper Summary:

This paper looks at empirically measuring neural network architecture expressivity by examining performance on a variety of complex datasets, measuring dataset complexity with algebraic topology. The paper first introduces the notion of topological equivalence for datasets -- a desirable measure to use as it is invariant to superficial differences such as rotation, translation and curvature. The definition of homology from algebraic topology can then be used as a robust measure of the "complexity" of a dataset. This notion of difficulty focuses roughly on determining the number of holes of dimension n (for varying n) there are in the dataset, with more holes roughly leading to a more complex connectivity pattern to learn. They provide a demonstration of this on two synthetic toy datasets in Figure 1, training two (very small -- 12 and 26 neuron) single hidden layer networks on these two datasets, where the smaller of the two networks is unable to learn the data distribution of the second dataset. These synthetic datasets have a well defined data distribution, and for an empirical sample of N points, a (standard) method of determining connectivity by growing epsilon balls around each datapoint in section 2.3.

The authors give a theoretical result on the importance of homology: if a binary classifier has support homology not equal to the homology of the underlying dataset, then there is at least one point that is misclassified by the classifier. Experiments are then performed with single hidden layer networks on synthetic datasets, and a phase transition is observed: if h_phase is the number of hidden units where the phase transition happens, and h' < h < h_phase, h' has higher error and takes longer to converge than h. Finally, the authors touch on computing homology of real datasets, albeit with a low dimensional projection (e.g. down to 3 dimensions for CIFAR-10).

Main Comments

The motivation to consider algebraic topology and dataset difficulty is interesting, but I think this method is ultimately ill suited and unable to be adapted to more complex and interesting settings. In particular, the majority of experiments and justification of this method comes from use on a low dimensional manifold with either known data distribution, or with a densely sampled manifold. (The authors look at using CIFAR-10, but project this down to 3 dimensions -- as current methods for persistent homology cannot scale -- which somewhat invalidates the goal of testing this out on real data.) This is an important and serious drawback because it seems unlikely that the method described in Figure 3 of determining the connectivity patterns of a dataset are likely to yield insightful results in a high dimensional space with very few datapoints (in comparison to 2^{dimension}), where distance between datapoints is unlikely to have any nice class related correspondence.

Furthermore, while part of the motivation of this paper is to use dataset complexity measured with topology to help select architectures, experiments demonstrating that this might be useful are very rudimentary. All experiments only look at single hidden layers, and the toy task in Figure 1 and in section 3.2.1 and Figure 5 use extremely small networks (hidden size 12-26). It's hard to be convinced that these results necessarily generalize even to other larger hidden layer models. On real datasets, exploring architectures does not seem to be done at all (Section 4).


Minor Comments
Some kind of typo in Thm 1? (for all f repeated twice)
Small typos (missing spaces) in related work and conclusion
How is h_phase determined? Empirically? (Or is there a construction?)

Review Summary:

This paper is not ready to be accepted.

---

> ### Author Response · Authors · 2017-12-18
> **Author Response**
>
> We thank the reviewer for their valuable feedback and useful comments, and in particular we intend on addressing several points therein.
>
> >>>>> This is an important and serious drawback because it seems unlikely that the method described in Figure 3 of determining the connectivity patterns of a dataset are likely to yield insightful results in a high dimensional space with very few datapoints (in comparison to 2^{dimension}), where distance between datapoints is unlikely to have any nice class related correspondence. <<<<<
>
> From a conceptual standpoint, there is no reason why persistent homology would be unable to determine the topology of a dataset in the high-dimensional setting. Specifically our method applies persistent homology directly to the individual classes of a dataset, and so the concern of distance sparsity invalidating class related correspondence does not apply. Furthermore, computation of persistent homology on high dimensional datasets via low-dimensional isometric embedding is equivalent to that on the original dataset; in other words, we present our methodology on CIFAR-10 as a template for how such computations can be done practically on real-world datasets without invalidating the integrity of the topological complexity computed. Relative to the latent space, the topological dimension of each individual class in CIFAR-10 is actually quite small in comparison to the ambient space
>
>
> The reviewer points out several concerns which we intend on addressing in a revision of the paper. First, the latent dimension given in the isometric embedding of CIFAR-10 is far too low in the first revision of the paper. Second, we intend on giving a more substantial characterization of larger hidden layer models as well as extending the empirical analysis to decision boundary (not just region) complexity. Lastly, we intend on taking results of computations on the various real world datasets given and suggesting and testing various architectures given by our characterization.

---

### Official Review · AnonReviewer1 · 2017-11-27
**A good idea, but undercooked**

**Rating:** 4
**Confidence:** 5

**Review:**

The authors propose to use the homology of the data as a measurement of the expressibility of a deep neural network. The paper is mostly experimental. The theoretical section (3.1) is only reciting existing theory (Bianchini et al.). Theorem 3.1 is not surprising either: it basically says spaces with different topologies differ at some parts.

As for the experiments, the idea is tested on synthetic and real data. On synthetic data, it is shown that the number of neurons of the network is correlated with the homology it can express. On real data, the tool of persistent homology is applied. It is observed that the data in the final layer do have non-trivial signal in terms of persistent homology.

I do like the general idea of the paper. It has great potentials. However, it is much undercooked. In particular, it could be improved as follows:

* 1) the main message of the paper is unclear to me.  Results observed in the synthetic experiments seem to be a confirmation of the known results by Bianchini et al.: the Betti number a network can express is linear to the number of hidden units, h, when the input dimension n is a constant.

To be convinced, I would like to see much stronger experimental evidence: Reporting results on a single layer network is unsettling. It is known that the network expressibility is highly related to the depth (Eldan & Shamir 2016). So what about networks with more layers? Is the stratification observation statistically significant? These experiments are possible for synthetic data.

* 2) The usage of persistent homology is not well justified. A major part of the paper is devoted to persistent homology. It is referred to as a robust computation of the homology and is used in the real data experiments. However, persistent homology itself was not originally invented to recover the homology of a fixed space. It was intended to discover homology groups at all different scales (in terms of the function value). Even with the celebrated stability theorem (Cohen-Steiner et al. 2007) and statistical guarantees (Chazal et al. 2015), the relationship between the Vietoris-Rips filtration persistent homology and the homology of the classifier region/boundary is not well established. To make a solid statement, I suggest authors look into the following papers

Homology and robustness of level and interlevel sets
P Bendich, H Edelsbrunner, D Morozov, A Patel, Homology, Homotopy and Applications 15 (1), 51-72, 2013

Herbert Edelsbrunner, Michael Kerber: Alexander Duality for Functions: the Persistent Behavior of Land and Water and Shore. Proceedings of the 28th Annual Symposium on Computational Geometry, pp. 249-258 (SoCG 2012)

There are also existing work on how the homology of a manifold or stratified space can be recovered using its samples. They could be useful. But the settings are different: in this problem, we have samples from the positive/negative regions, rather than the classification boundary.

Finally, the gap in concepts carries to experiments. When persistent homology of different real data are reported. It is unclear how they reflect the actually topology of the classification region/boundary. There are also significant amount of approximation due to the natural computational limitation of persistent homology. In particular, LLE and subsampling are used for the computation. These methods can significantly hurt persistent homology computation. A much more proper way is via the sparsification approach.

SimBa: An Efficient Tool for Approximating Rips-Filtration Persistence via Simplicial Batch-Collapse
T. K. Dey, D. Shi and Y. Wang. Euro. Symp. Algorithms (ESA) 2016, 35:1--35:16

* 3) Finally, to support the main thesis, it is crucial to show that the topological measure is revealing information existing ones do not. Some baseline methods such as other geometric information (e.g., volume and curvature) are quite necessary.

* 4) Important papers about persistent homology in learning could be cited:

Using persistent homology in deep convolutional neural network:

Deep Learning with Topological Signatures
C. Hofer, R. Kwitt, M. Niethammer and A. Uhl, NIPS 2017

Using persistent homology as kernels:

Sliced Wasserstein Kernel for Persistence Diagrams
Mathieu Carrière, Marco Cuturi, Steve Oudot, ICML 2017.

* 5) Minor comments:

Small typos here and there: y axis label of Fig 5, conclusion section.

---

> ### Author Response · Authors · 2017-12-18
> **Author Response (pt. 1)**
>
> We thank the reviewer for the valuable feedback and numerous suggestions for improvement. We will address and further inquire into each point individually.
>
>
> >>>>>The authors propose to use the homology of the data as a measurement of the expressibility of a deep neural network. The paper is mostly experimental. The theoretical section (3.1) is only reciting existing theory (Bianchini et al.).  Theorem 3.1 is not surprising either: it basically says spaces with different topologies differ at some parts. <<<<<
>
> While we agree that those with some background in topology (including the authors) would view Theorem 3.1 as following trivially, the background and motivation of this paper is aimed towards the deep learning community at large. To illustrate that topology is at least a meaningful minimality condition for the expressivity of neural architectures, we think that the proposition is an important conceptual stepping stone.  There’s a tradeoff between highlighting conceptual importance and rigorous underpinnings to readers unversed in topology and acknowledging simplicity and triviality to those who are. In this case, we felt the former to be more useful to the general audience.
>
>
> >>>>> * 1) the main message of the paper is unclear to me.  Results observed in the synthetic experiments seem to be a confirmation of the known results by Bianchini et al.: the Betti number a network can express is linear to the number of hidden units, h, when the input dimension n is a constant.  <<<<<
>
> In this work, we wish to give an exact empirical characterization of each individual Betti number, demonstrate that topological phenomena during training with stochastic gradient descent, and most importantly show that there is potential to use topology as a measure of data complexity to give a reasonable range of trial architectures for architecture search or human-aided architecture selection.
>
> We would like to highlight that in Bianchini’s work, it is only shown that there is an exact linear relationship on the sum of Betti numbers for the arctan activation. Furthermore, the bounds given in Bianchini’s work are a result of landmark work on bounding the sum of Betti numbers of sub-algebraic sets (Basu, 1999), which then led to the Pfefferian bounds used. From a theoretical perspective (Basu, 1999), actually gives a bounding method for individual Betti numbers in the long exact Mayer-Vietoris sequence, but subsumes those methods by a bound on the sum.
>
> In the context of architecture selection, bounding each individual Betti number is a crucial next step in developing the pipeline from persistent homology to minimal architecture. Furthermore, we feel an illustration of the empirical study of topological expressivity is essential in gauging the practical utility of these bounds.
>
> >>>>> To be convinced, I would like to see much stronger experimental evidence: Reporting results on a single layer network is unsettling. It is known that the network expressibility is highly related to the depth (Eldan & Shamir 2016). So what about networks with more layers? Is the stratification observation statistically significant? These experiments are possible for synthetic data. <<<<<
>
> We thank the reviewer for this suggestion. We hope to complete experiments with more layers and different activation functions in the final draft of the manuscript. As to the stratification observation, we will add $p$ values for appropriate hypothesis tests.

---

> > ### Author Response · Authors · 2017-12-18
> > **Author Response (pt. 2)**
> >
> > >>>>> Persistent homology itself was not originally invented to recover the homology of a fixed space. It was intended to discover homology groups at all different scales (in terms of the function value). <<<<<
> >
> > In a morse theoretic sense, the reviewer’s definition of persistent homology is absolutely correct. However in the seminal work of Zomorodian and Carlsson, persistence homology is motivated in three contexts, recovering the static homology of a space from its point cloud, recovering the static homology of a space from a point cloud sampled from a distribution concentrated on a static space, and the reviewer’s given motivation, recovering homology of submanifolds according to excursion sets of a Morse function. It’s fair to say our use of persistent homology falls squarely in the first two categories.
> >
> > For example,  “Persistence complexes arise naturally whenever one is attempting to study topological invariants of a space computationally. Often, our knowledge of this space is limited and imprecise. Consequently, we must utilize a multiscale approach to capture the connectivity of the space, giving us a persistence complex.” (Zomorodian and Carlsson, 2005)
> >
> > In particular, immediately following this statement they contextualize persistent homology in the context of trying to estimate the static topological invariants of a fixed space X from point cloud samples:  “Example 1.1 (point cloud data) Suppose we are given a finite set of points X from a subspace Y ∈ R^n. We call X point cloud data or PCD for short. It is reasonable to believe that if the sampling is dense enough, we should be able to compute the topological invariants of Y directly from the PCD. To do so, we may either compute the Cech ˇ complex, or approximate it via a Rips complex [15]. [...].” (Zomorodian and Carlsson, 2005)
> >
> > >>>>> The usage of persistent homology is not well justified. [...] Even with the celebrated stability theorem (Cohen-Steiner et al. 2007) and statistical guarantees (Chazal et al. 2015), the relationship between the Vietoris-Rips filtration persistent homology and the homology of the classifier region/boundary is not well established. <<<<<
> >
> > The one-versus-all setting studied in this work is limited of course in the case where the geometric complexity of the decision boundary is simpler than that of the individual classes, themselves. However, for the case of finding sufficiently powerful architectures we feel that this is a good starting point, as in the worst case, the decision boundary will inherit the complexity of the individual classes. Moreover, in the generative and unsupervised setting, capturing the support of the distributions of each class is a necessity, and therefore the use of persistent homology on individual classes directly applies.
> >
> > On the other hand, there has been recent work on directly building simplicial complexes between multiple classes which empirically characterizes the decision boundary sufficiently to aid in optimal kernel selection (Varshney and Ramamurthy, 2015). We will provide a comparison between the foregoing method and ours for characterizing learned decision boundary topology in a final version of this work.
> >
> >
> >
> > >>>>> 3) Finally, to support the main thesis, it is crucial to show that the topological measure is revealing information existing ones do not. Some baseline methods such as other geometric information (e.g., volume and curvature) are quite necessary. <<<<<
> >
> > To our knowledge, our work gives the first relationship between a computable measure of data complexity and the learnability of architectures with respect to that measure. If the reviewer knows of any other such similar baselines to which we can compare, we would greatly appreciate if references could be provided.  From a theoretical perspective, it is clear that topology reveals geometric information that volume and curvature do not, so we feel that if a baseline is to be made, then the comparison would be strictly empirical. However, without other related literature relating geometric information to architecture optimality, we believe the theoretical results of Bianchini et al. are a good baseline.
> >
> >
> > #################
> > References:
> >
> > Basu, S., 1996, July. On bounding the Betti numbers and computing the Euler characteristic of semi-algebraic sets. In Proceedings of the twenty-eighth annual ACM symposium on Theory of computing (pp. 408-417). ACM.
> >
> > Zomorodian A, Carlsson G. Computing persistent homology. Discrete & Computational Geometry. 2005 Feb 1;33(2):249-74.
> >
> > Varshney, Kush R., and Karthikeyan Natesan Ramamurthy. "Persistent topology of decision boundaries." Acoustics, Speech and Signal Processing (ICASSP), 2015 IEEE International Conference on. IEEE, 2015.

---

### Official Review · AnonReviewer3 · 2017-11-28
**The paper aims to connect topology of data to that of decision super-level sets and boundaries, as a new characterization of the capacity of neural networks. The empirical study is very inspiring, yet the paper can be improved.**

**Rating:** 4
**Confidence:** 5

**Review:**

General comments:

The paper is largely inspired by a recent work of Bianchini et al. (2014) on upper bounds of Betti number sums for decision super-level sets of neural networks in different architectures. It explores empirically the relations between Betti numbers of input data and hidden unit complexity in a single hidden layer neural network, in a purpose of finding closer connections on topological complexity or expressibility of neural networks.

They report the phenomenon of phase transition or turning points in training error as the number of hidden neurons changes in their experiment. The phenomenon of turning points has been observed in many experiments, where usually researchers investigate it through the critical points of training loss such as local optimality and/or saddle points. For the first time, the paper connects the phenomenon with topological complexity of input data and decision super-level sets, as well as number of hidden units, which is inspiring.

However, a closer look at the experimental study finds some inconsistencies or incompleteness which deserves further investigations. The following are some examples.

The paper tries to identify a phase transition in number of hidden units, h_phase(D_2) = 10 from the third panel of Figure 4. However, when h=12 hidden units, the curve is above h=10 rather than below it in expectation. Why does the order of errors disagree with the order of architectures if the number of hidden neurons is larger then h_phase?

The author conjecture that if b0 = m, then m+2 hidden neurons are sufficient to get 0 training error.
But the second panel of fig4 seems to be a counterexample of the conjecture. In fact h_phase(D_0 of b_0=2)=4 and h_phase (D_1 of b_0 = 3) = 6, as pointed out by the paper, has a mismatch on such a numerical conjecture.

In Figure 5, the paper seems to relate the homological complexities of data to the hidden dimensionality in terms of zero training error. What are the relations between the homological complexities of data and homological complexities of decision super-level sets of neural networks in training? Is there any correspondence between them in terms of topological transitions.

The study is restricted to 2-dimensional synthetic datasets. Although they applied topological tools to low-dimensional projection of some real data, it's purely topological data analysis. They didn't show any connection with the training or learning of neural networks. So this part is just preliminary but incomplete to the main topic of the paper.

The authors need to provide more details about their method and experiments. For example, The author didn't show from which example fig6 is generated. For other figures appended at the end of the paper, there should also be detailed descriptions of the underlying experiments.


Some Details:

Lines in fig4 are difficult to read, there are too many similar colors. Axis labels are also missing.

In fig5, the (4, 0)-item appears twice, but they are different. They should be the same, but they are not.
Any mistake here?

Fig6(a) has the same problem as fig4. Besides, the use of different x-axis makes it difficult to compare with fig4.
fig6(b) needs a color bar to indicate the values of correlations.

Some typos, e.g. Page 9 Line 2, 'ofPoole' should be 'of Poole'; Line 8, 'practical connectio nbetween' should be 'practical connection between'; line 3 in the 4th paragraph of page 9, 'the their are' seems to be 'there are'. Spell check is recommended before final version.

---

> ### Author Response · Authors · 2017-12-18
> **Author Response**
>
> We would like thank the reviewer for their insightful comments and perspective. In particular, there were several concerns as to the completeness and consistency of the analysis given in the submission that we would like to address and correct.
>
> Over the course of developing the manuscript several figures were rendered at different levels of completion of the main experiments. The different (sub)figures contain an accurate reflection of the data collected at their time of rendering, but as a result of the substantial duration of some of these experiments our oversight has led to the mentioned inconsistencies. In order to maintain a general level of transparency and rigor in our work, we will promptly rerun the major experiments of the paper, publish all of our code on Github, rerender the given figures, and provide a full addendum to the paper which exactly indicates our experimental methodology.
>
>
>
> >>>>> The author conjecture that if b0 = m, then m+2 hidden neurons are sufficient to get 0 training error. But the second panel of fig4 seems to be a counterexample of the conjecture. In fact h_phase(D_0 of b_0=2)=4 and h_phase (D_1 of b_0 = 3) = 6, as pointed out by the paper, has a mismatch on such a numerical conjecture.   <<<<<
>
> In our experiment we intended to present empirically driven conjectures on a lower bound for the homological expressivity of networks. We thank the reviewer for pointing out a flaw in the statement of conjecture: in particular, the statement should be, there exists a dataset $\mathcal{D}$ with $H_0(\scriptd)$ = \mathbb{Z}^m$ so that a single hidden layer neural network with $h =m +2$ hidden units converges to zero error on $\mathcal{D}$. In the case of the given dataset in the second panel of figure 4, although it satisfies this homological property, it indeed needs  6 hidden units. However, the following is a construction of a dataset which satisfies the existence claim in the conjecture. Take $D$ such that 3 horizontally separated columns of data points extend vertically in $\mathbb{R}^2, then the superposition of two “cylindrical” bumps and one half space suffice to cover this dataset. We amended our conjecture to include similar such constructions. Despite the existence of datasets satisfying the foregoing constraints, we intend on changing the statement to one of the existence of neural networks that express the given homology and not those which train to zero error.
>
>
> >>>>> In fig5, the (4, 0)-item appears twice, but they are different.  <<<<<
>
> This was an oversight in rendering the second, third, and fourth panel at a point in the experiment before its completion. As aforementioned, we will rerun the main experiments and update the paper with the proper renderings.
>
>
> >>>> The paper tries to identify a phase transition in number of hidden units, h_phase(D_2) = 10 from the third panel of Figure 4. However, when h=12 hidden units, the curve is above h=10 rather than below it in expectation. Why does the order of errors disagree with the order of architectures if the number of hidden neurons is larger then h_phase? <<<<<
>
> Although we did not reserve enough space in the manuscript to mention this, this isn’t just an inconsistency of Figure 4, but something we noticed ubiquitously as we increased the homological complexity of the dataset; that before the phase transition point there appears to be a strict order of architectures, after this it seems to give out to noise. In order to further investigate this phenomena, we intend on increasing the number of networks trained and the variation in the datasets given in a final draft of the work.
>
>
> >>>>> In Figure 5, the paper seems to relate the homological complexities of data to the hidden dimensionality in terms of zero training error. What are the relations between the homological complexities of data and homological complexities of decision super-level sets of neural networks in training? Is there any correspondence between them in terms of topological transitions.  <<<<<
>
> We should add to the paper that Figure 5 gives the final testing error of neural networks with respect to homological complexity of datasets. The testing error gives a full representation of a network to express the homological complexity of the dataset in the homology of its own decision boundary; that is, at the end of training the networks express the homology of the given dataset.
>
>
> >>>>> Although they applied topological tools to low-dimensional projection of some real data, it's purely topological data analysis. <<<<<
>
> We agree that the analysis of real data is incomplete as network architectures with networks at a predicted h_phase point were not tested, and we fully intend on completing this analysis in the final draft of the work. There is however some merit in applying TDA to standard benchmark datasets in order to demonstrate the existence of non-trivial topological features therein.

---

### Decision · Program_Chairs · 2018-01-29
**ICLR 2018 Conference Acceptance Decision**

**Decision:**

Reject

**Comment:**

This paper attempts to connect the expressivity of neural networks with a measure of topological complexity. The authors present some empirical results on simplified datasets.
All reviewers agreed that this is an intriguing line of research, but that the current manuscript is still presenting preliminary results, and that further work is needed before it can be published.